# Unveiling a Novel Role of Cdc42 in Pyruvate Metabolism Pathway to Mediate Insecticidal Activity of *Beauveria bassiana*

**DOI:** 10.3390/jof8040394

**Published:** 2022-04-12

**Authors:** Yi Guan, Donghuang Wang, Xiaofeng Lin, Xin Li, Chao Lv, Dingyi Wang, Longbin Zhang

**Affiliations:** 1Fujian Key Laboratory of Marine Enzyme Engineering, College of Biological Science and Engineering, Fuzhou University, Fuzhou 350116, China; wangdonghuang_2022@163.com (D.W.); 200820080@fzu.edu.cn (X.L.); n190827020@fzu.edu.cn (X.L.); lchazh@163.com (C.L.); 2School of Geographical Sciences, Fujian Normal University, Fuzhou 350007, China; albertwdy@fjnu.edu.cn

**Keywords:** entomopathogenic fungi, cell cycle, gene expression and regulation, pyruvate metabolism, TCA cycle, beauvericin, Pr1 family proteases, virulence

## Abstract

The small GTPase Cdc42 acts as a molecular switch essential for cell cycles and polar growth in model yeast, but has not been explored in *Beaurveria bassiana*, an insect-pathogenic fungus serving as a main source of fungal formulations against arthropod pests. Here, we show the indispensability of Cdc42 for fungal insecticidal activity. Deletion of *cdc42* in *B. bassiana* resulted in a great loss of virulence to *Galleria mellonella*, a model insect, via normal cuticle infection as well as defects in conidial germination, radial growth, aerial conidiation, and conidial tolerance to heat and UVB irradiation. The deleted mutant’s hyphae formed fewer or more septa and produced unicellular blastospores with disturbed cell cycles under submerged-culture conditions. Transcriptomic analysis revealed differential expression of 746 genes and dysregulation of pyruvate metabolism and related pathways, which were validated by marked changes in intracellular pyruvate content, ATP content, related enzyme activities, and in extracellular beauvericin content and Pr1 protease activity vital for fungal virulence. These findings uncover a novel role for Cdc42 in the pathways of pyruvate metabolism and the pyruvate-involved tricarboxylic acid cycle (TCA cycle) and a linkage of the novel role with its indispensability for the biological control potential of *B. bassiana* against arthropod pests.

## 1. Introduction

*Beauveria bassiana* (Hypocreales: Cordycipitaceae) is a well-known insect-pathogenic fungus that serves as a main source of wide-spectrum fungal insecticides [1]. The fungal-infection cycle usually begins from conidial adherence to insect cuticle, followed by conidial germination and hyphal growth for penetration through the host cuticle [2]. In the penetration process, multiple families of extracellular enzymes contribute to cuticle degradation, such as proteases, chitinases, and lipases [3,4,5,6,7]. After hyphal invasion into the insect body, the hyphae turn into unicellular hyphal bodies (i.e., blastospores) to proliferate by yeast-like budding in insect hemocoel until host death from mummification and/or production of secondary metabolites with insecticidal activities, such as beauvericin, which is considered important for lethal action [8,9]. In the dying host, hyphal bodies turn back into septate hyphae to penetrate the host cuticle for outgrowth and conidiation on the surfaces of insect cadavers [10,11,12]. The whole infection cycle of *B. bassiana* comprises an array of cellular processes and events associated with the fungal potential against insect pests. In the past decade, hundreds of genes in *B. bassiana* have been functionally characterized in association with various aspects of the fungal biocontrol potential, leading to the identification of numerous candidate genes highly potential for use in the genetic improvement of fungal virulence and stress tolerance [13,14,15,16,17]. In these studies, the agrobacterium-mediated transformation was widely applied for gene deletion or complementary-strain construction in *B. bassiana* for its simple, highly efficient, and reliable traits. A phosphinothricin acetyltransferase (*bar*) gene was induced in this method as a selectable marker [18].

Cdc42 (cell-division cycle 42) is a classical small GTPase classified to the Rho family and was first discovered by screening yeast mutants defective in bud formation [19,20,21]. Cdc42 has since been studied intensively in model yeast and is well-known as a regulator of the cell cycle/division and important cellular events, including in particular polarized growth and septin-ring dynamics [19,22,23,24,25]. Aside from an involvement in polarization, Cdc42 also serves as a pheromone-signaling factor [26,27,28,29,30,31,32]. In filamentous fungi, Cdc42 also plays a conserved role in the establishment of cell polarity. As examples, Cdc42 is required for polarity establishment or correct cell polarization in *Aspergillus nidulans* [33], *Candida albicans* [34], and *Penicillium marneffei* [35]. In addition, Cdc42 has proved important for fungal virulence, which was attenuated by knockout mutations of *cdc42* in *Botrytis cinerea*, *Claviceps purpurea*, *Colletotrichum gloeosporioides*, and *Nomuraea rileyi* [36,37,38,39]. The previous studies demonstrate not only a conserved role for Cdc42 in the fungal cell cycle and polar growth but also its special role in filamentous fungal adaptation to host infection. However, orthologous Cdc42 remains unexplored yet in *B. bassiana*, making it unclear whether it has a special role in the fungal insect-pathogenic lifecycle.

Transcriptomic analysis is a powerful tool to reveal changes in genome-wide gene expression associated with the fungal lifecycle, and has been widely used to explain altered phenotypes caused by the disruption of a specific gene, such as the aberrance of sucrose utilization caused by the knockout mutation of *sur7* in *B. bassiana* [40]. In this study, the biological function of Cdc42 in *B. bassiana* was investigated by phenotypic and transcriptomic analyses of its knockout mutant (Δ*cdc42*) in parallel with the parental wild-type strain, with an emphasis placed on its impact on fungal biocontrol potential against insect pests. We found an essential role of Cdc42 in not only the fungal cell cycle/division but also the fungal insect-pathogenic lifestyle. Our transcriptomic analysis also revealed its regulatory role in the expression of gene clusters involved in pyruvate metabolism and energy production.

## 2. Materials and Methods

### 2.1. Microbial Strains and Culture Conditions

The wild-type strain *B. bassiana* ARSEF 2860 (WT herein) was stored at RW Holley Center for Agriculture and Health, Ithaca, New York, NY, USA. Its genome data has been published in GenBank (Accession No. of ADAH00000000). In the past decade, *B. bassiana* ARSEF 2860 has been used as a model strain for numerous gene-function studies [10,11,12,13,14,15,16,17]. WT and its mutants were cultured on Sabouraud dextrose agar (SDAY; 4% glucose, 1% peptone and 1.5% agar plus 1% yeast extract) for fungal growth or in SDBY (i.e., agar-free SDAY) for liquid culture at 25 °C in a light/dark cycle of 12:12 h. *Escherichia coli* Top10 and *E. coli* DH5a from Invitrogen (Shanghai, China) were cultivated for vector propagation at 37 °C in Luria–Bertani medium plus ampicillin (100 mg/mL) or kanamycin (50 mg/mL). *Agrobacterium tumefaciens* AGL-1 incubated in the YEB medium [18] was used as a T-DNA donor for fungal transformation.

### 2.2. Recognition and Bioinformatic Analysis of Cdc42 in B. Bassiana

The full-length sequence of *S. cerevisiae* Cdc42 (NCBI code: QHB10383) was used as a query to locate Cdc42 in the *B. bassiana* genome [41]. The coding sequence of the located Cdc42 (NCBI code: EJP68839) was amplified from the WT DNA with a pair of primers (Appendix A) and sequenced for verification at Invitrogen. The protein sequence deduced from the verified nucleotide sequence was subjected to online blast analysis for its structural features and aligned with the Cdc42 sequence of *A. nidulans* using the SMART program. Phylogenetic analysis was then performed for the Cdc42 homologues in several fungi using a neighbor-joining method in MEGA7 software.

### 2.3. Generation of cdc42 Mutants

The backbone plasmids p0380-*bar* and p0380-*sur*-gateway [42] were used to construct plasmids for *cdc42* deletion and complementation. Briefly, the 5’ and 3´ fragments (1428 and 1819 bp, respectively) of *cdc42* comprising partial coding and flanking regions were amplified from the WT DNA with paired primers (Appendix A) and inserted into p0380-*bar* at the XmaI/BamHI and XbaI/SpeI sites, respectively, forming p0380-5´*cdc42*-bar-3´*cdc42*. The full-length sequence of *cdc42* and its flanking regions (3501 bp in total) were amplified from the WT DNA and inserted into the p0380-*sur*-gateway to exchange for the gateway fragment under the action of Gateway BP Clonase^TM^ II Enzyme Mix (Invitrogen), yielding p0380-sur-*cdc42* vectoring the sur marker. The two constructed plasmids were propagated in *E. coli* Top10 and *E. coli* DH5α and transformed into the WT and the Δ*cdc42* mutant via *Agrobacterium*-mediated transformation [18], respectively. Putative mutants were screened in terms of the *bar* resistance to phosphinothricin (200 μg/mL) or the sur resistance to chlorimuron ethyl (15 μg/mL) in a selective medium. The expected recombination events (Appendix A) were verified by PCR (Appendix A) with paired primers (Appendix A). Positive Δ*cdc42* mutant and its control strains (parental WT and Δ*cdc42*::*cdc42*) were used in phenotypic experiments including three independent replicates.

### 2.4. Phenotypic Experiments

The growth rate of each strain was initiated by spotting 1 μL aliquots of a 10^6^ conidia/mL suspension on SDAY plates. After an 8-day incubation at 25 °C and 12:12 h (L:D), the mean diameter of each colony was estimated as an index of growth rate.

Cultures used for assessment of conidial capacity were initiated by spreading 100 μL of a 10^7^ conidia/mL suspension per SDAY plate and incubated for 7 d at 25 °C and 12:12 h (L:D). From day 3 onwards, three plugs (4 mm diameter) were bored daily from each plate culture. The conidia on each plug were released into 1 mL of 0.02% Tween80 via thorough vibration. The conidial concentration in the suspension was assessed using a hemocytometer and converted to the number of conidia per square centimeter of plate culture.

Conidia collected from each of the cultures were suspended in a germination broth (2% sucrose and 0.5% peptone in 0.02% Tween 80), and three aliquots (standardized to 10^6^ conidia/mL) were shaken by 180 rpm at 25 °C for 24 h. Germination percentage in each aliquot was determined every 2 h during the incubation using a hemocytometer and the trend of germination over the time was subjected to modeling analysis, yielding an estimate of median germination time (GT_50_).

Conidial thermotolerance and UV-B resistance of each strain were assayed as described previously [43,44]. Briefly, three samples of conidia were exposed to a hot-water bath at 45 °C for up to 120 min or the UV-B irradiation of weighted wavelength of 312 nm at gradient doses from 0 to 0.5 J/cm^2^ in Bio-Sun^++^ chamber (Vilber Lourmat, Marne-la-Vallée, France). After exposure, each sample was incubated for 24 h at 25 °C under the conditions of saturated humidity, followed by germination percentage determined under a microscope. Conidial thermotolerance and UV-B resistance were estimated as LT_50_ (min) and LD_50_ (J/cm^2^) by modeling analyses of the conidial survival trends over the gradient intensities of the two stresses, respectively.

The virulence of each strain was bioassayed through normal cuticle infection of *Galleria mellonella* larvae (instar V) from a vendor (Da Mai Chong Insectaries, Wuxi, Jiangsu, China). Briefly, three groups of 30–40 larvae per strain were separately immersed in 40 mL aliquots of 0.02% Tween 80 (control) or a 10^7^ conidia/mL suspension. All treated groups were maintained in Petri dishes (15 cm diameter) at 25 °C for 8 d and monitored daily for mortality/survival records. The resultant time-mortality trend from each group was subjected to probit analysis for the estimation of an LT_50_ (day) as a virulence index.

During the period of bioassay, hemolymph samples were taken from the larvae surviving the normal infection for 5 d and observed under a microscope to reveal a status of fungal proliferation in vivo. The concentration of hyphal bodies from each of three samples per larvae (three larvae per strain) was assessed with hemocytometer.

### 2.5. Examination of Cell Cycle and Division

The 50 mL aliquots of a 10^5^ conidia/mL suspension in SDBY were incubated at 25 °C for 3 d on a shaking bed (150 rpm). Hyphal samples taken from the cultures were stained with the cell-wall-specific dye calcofluor white and visualized with laser scanning confocal microscopy (LSCM) to reveal septum pattern of each strain.

Thin-wall unicellular blastospores collected from the cultures were stained with the DNA-specific dye propidium iodide, followed by fluorescence-activated cell-sorter (FACS) analysis (Instrument SN: AN12056 Software Version: CXP 2.2) to assess G_1_/G_0_, G_2_/M and S phases of cell cycle in 2 × 10^4^ stained blastospores per sample (three samples per strain with an argon laser at the excitation/emission wavelengths of 488/530 (±15) nm in the flow cytometer FC 500 MCL (Beckman Coulter, CA, USA).

### 2.6. Transcriptomic Analysis

The Δ*cdc42* and WT strains were cultured on SDAY plates for 5 d at the optimal regime. The resultant cultures (three replicates per strain) were sent to Majorbio (Shanghai, China) for transcriptomic analysis. The RNA-seq transcriptome library was consctructed following TruSeq^TM^ RNA sample preparation Kit from Illumina (San Diego, CA, USA). Double-stranded cDNA was synthesized using a SuperScript double-stranded cDNA synthesis kit (Invitrogen, CA, USA) with random hexamer primers (Illumina). The paired-end RNA-seq library was sequenced on Illumina HiSeq xten/NovaSeq 6000 sequencer (2 × 150 bp read length). The level of each transcript was calculated according to the method of transcripts per million reads (TPM). The abundance of each gene was quantified with RSEM [45] at http://deweylab.biostat.wisc.edu/rsem/ (accessed on 27 May 2021). Differentially expressed genes (DEGs) were identified from the Δ*cdc42* versus WT transcriptome based on fold change >1 (upregulated) or <0.5 (downregulated) at the significance level of *p* ≤ 0.05. All identified DEGs were subject to gene ontology (GO) analysis for enrichment to GO classes and terms and to Kyoto Encyclopedia of Genes and Genomes (KEGG) analysis for enrichment to KEGG pathways at the significant level of *p* ≤ 0.05. GO and KEGG analyses were carried out with the online programs Goatools (https://github.com (accessed on 27 May 2021)/tanghaibao/Goatools) and KOBAS (http://kobas.cbi.pku.edu.cn/home.do, accessed on 27 May 2021), respectively.

### 2.7. Assessments of Protein and ATP Levels Associated with TCA Cycle

Due to an important role of Cdc42 revealed by transcriptomic analysis in the fungal “tricarboxylic acid cycle” (TCA cycle), related enzyme activities and ATP levels were assessed from submerged cultures of WT and Δ*cdc42* strains. Briefly, 50 μL aliquots of a 10^5^ conidia/mL suspension in SDBY were incubated at 25 °C for 5 d on the shaking bed. The supernatant of each culture was used to assay the activities (U/mg) of citrate synthase (CS) and fumarate dehydrogenase (FDH) using the corresponding enzyme kits (Solarbio Science & Technology, Beijing, China). Total protein was determined in Bradford assay.

To assess pyruvate and ATP contents, cells from each of the 5-day-old SDBY cultures were ground with liquid nitrogen, resuspended in extraction buffer, and followed by a 30 min ice bath. After a 10 min centrifugation at 10,000 rpm, the supernatant was mixed with 500 μL chloroform via vibration, followed by a 3 min centrifugation. The supernatant was used to assess the contents (μM/g) of pyruvate and ATP following the users’ guides of the corresponding detection kits from Solarbio.

### 2.8. Assessments of Beauvericin Level and Pr1 Family Protease Activity

For in-depth insight into the fungal virulence greatly attenuated in the absence of *cdc42*, 5-day-old SDBY cultures were generated as aforementioned. The supernatant was collected by a 10 min centrifugation at 4 °C and mixed with an equal volume of ethyl acetate for extraction of beauvericin at 4 °C overnight. The extracted supernatant was dried by nitrogen gas blowing, concentrated 10-fold with 80% methanol, and filtered through a 0.22 μm filter. The content of beauvericin in each sample was determined by high-performance liquid chromatography (HPLC). HPLC was conducted with Diamonsil C18 (250 mm × 4.6 mm, 5 μm) as chromatographic column and methanol: with water (*v*/*v* 70:30) gradient-elution mobile phase at a flow rate of 0.9 mL/min, and an injection volume of 10 μL. The peak wavelength of beauvericin was detected at 200 nm for a presence of ~10 min. Three biological replicates were included in the assay.

The total activity of Pr1 family proteases required for cuticle degradation during normal infection [7] was quantified directly from the supernatant of each 5-day-old SDBY culture as described previously [7,46]. Briefly, azocasein (Sigma, Shanghai, China) was dissolved in 50 mM Tris HCl (pH 8.0) to final concentration with 5 mg/mL. Every 100 μL of azocasein solution was mixed with 100 μL of supernatant (crude enzyme sample), followed by a 60 min incubation at 37 °C, and terminating the reaction by adding 400 μL of 10% (*w*/*v*) trichloroacetic acid. After a 4 min centrifugation, the supernatant was then transferred to 700 μL of 525 mM NaOH for reading optical density at 442 nm (OD_442_). One unit of activity was defined as an enzyme amount for an increase in the OD value by 0.01 after a 60 min reaction of each extract versus control.

## 3. Results

### 3.1. Recognition of Orthologous Cdc42 in B. Bassiana

The Cdc42 ortholog located in the *B. bassiana* genome [18] through blast analysis with the query sequence of yeast Cdc42 consists of 242 amino acids with pI/Mw of 8.44/27.15 kDa. It shares 99% sequence identity with the orthologs found in the filamentous fungi *Cordyceps militaris* and *Talaromyces marneffei* (Appendix A). Revealed by conserved domain analysis, Cdc42 orthologs in *B. bassiana* and *Aspergillus nidulans* feature five conserved G boxes (G1–G5) typical for the Ras superfamily, two switch domains, and a CAAX residue at C-termini (Appendix A).

### 3.2. Impact of cdc42 Deletion on Radial Growth, Conidiation, and Conidial Quality

Several phenotypes associated with the asexual cycle in vitro of *B. bassiana* were compared between the Δ*cdc42* mutant and its control strains. The Δ*cdc42* mutant’s radial growth initiated with ~10^3^ conidia on SDAY plates was markedly reduced by 33% in comparison to the WT’s colonies incubated for 7 d at the optimal regime of 25 °C and L:D 12:12 (Figure 1A). Despite insignificant changes at the earlier stages of conidiation, the mutant’s conidial yields on average decreased by 36% and 41% on days 6 and 7 after the SDAY cultures were initiated by spreading 100 μL of a 10^7^ conidia/mL suspension *per capita* (Figure 1B). The quality of the mutant’s conidia was also compromised, as indicated by a 27% increase in median germination time (GT_50_) at optimal 25 °C (Figure 1C), a 16% decrease in tolerance to a wet-heat stress at 45 °C (Figure 1D) and a 21% reduction in resistance to UV-B irradiation (Figure 1E).

The phenotypic changes observed in the Δ*cdc42* mutant were well restored by targeted gene complementation. These data indicated an important role of *cdc42* in the asexual cycle in vitro of *B. bassiana*.

### 3.3. Impact of cdc42 Deletion on Hyphal Septum Formation and Morphology

In multicellular fungi, hyphal elongation is under the control of the cell cycle, in which daughter cells are generated from the compartmentalization of parent cells by forming septa. Despite a longer time required for conidial germination, the Δ*cdc42* mutant showed an inconspicuous change in polar growth of germ tubes on plates (data not shown). Revealed by FACS analysis, cell-cycle phases of blastospores produced in the 3-day-old SDBY cultures of the Δ*cdc42* mutant were disturbed in comparison to the counterparts of the control strains, including significantly prolonged G_1_/G_0_, shortened G_2_/M, and only a residual S phase (Figure 2A).

Compared to the control strains, moreover, the Δ*cdc42* mutant displayed two opposite types of abnormal septum patterns under the submerged-culture conditions (Figure 2A). In the mutant’s culture, some hyphae had elongated cells with only one to two septa with no conspicuous change in morphology, while other hyphae increased largely in thickness, formed many more septa, and hence comprised short stout cells. These observations implicated an important role for *cdc42* in the fungal cell cycle, division, and hyphal septation.

### 3.4. Indispensability of cdc42 for Fungal Insect Pathogencity

In the standardized bioassays, normal cuticle infection of two control strains by the topical application of a 10^7^ conidia/mL suspension to *G. mellonella* larvae resulted in 100% mortality within 10 d and a mean (±SD) LT_50_ of 4.8 (±0.2) d (Figure 3A). In contrast, most larvae remained alive 15 d after infection by the Δ*cdc42* mutant, resulting in its LT_50_ being drastically prolonged to 16.9 (±1.7) d. This highlighted an indispensability of *cdc42* for the insect pathogenicity and virulence of *B. bassiana* through the normal infection.

Next, hemolymph samples taken from surviving larvae were examined to reveal a status of proliferation in vivo by yeast-like budding. As a result, hyphal bodies were observed in the samples of the larvae after a 5 d infection by either control strain, but were hardly observed in the larvae infected by the Δ*cdc42* mutant (Figure 3B). The concentrations of hyphal bodies formed by the mutant in the samples were averagely reduced by 60% in comparison to the WT’s concentrations, and the reduction was largely restored in the complementation strain (Figure 3C). These observations indicated an important role of c*dc42* in the fungal proliferation by yeast-like budding to facilitate mycosis development and host death.

### 3.5. Transcriptomic Insight into an Indispensability of cdc42 for Fungal Insect Pathogencity

Up to 746 DEGs (up/down ratio: 495:251) were identified from the transcriptome of the Δ*cdc42* mutant relative to the WT strain (Figure 4A, Appendix A). Revealed by GO analysis (Figure 4B), the deletion of *cdc42* exerted a profound effect on the cellular component (692 DEGs enriched to seven GO terms), biological process (956 DEGs enriched to nine GO terms), and molecular function (608 DEGs enriched to four GO terms). The enriched GO terms mainly included the components of membrane/membrane part; cell/cell part and organelle; the metabolic, single-organism and cellular processes; and the catalytic, binding and transporter activities. The KEGG analysis resulted in the enrichment of 60 DEGs to 12 pathways at the significant level *p* < 0.05 (Figure 4C). Of those, the pyruvate metabolism pathway was enriched at the most significant level of *p* < 0.001, followed by glycine, serine, and threonine metabolism. The enriched pathways were mostly involved in carbohydrate metabolism (pyruvate metabolism, glycolysis/gluconeogenesis, ascorbate and aldarate metabolism, and inositol phosphate metabolism), amino-acid metabolism (glycine, serine, and threonine metabolism; arginine biosynthesis; and valine, leucine, and isoleucine biosynthesis), lipid metabolism (glycerophospholipid metabolism, ether lipid metabolism, and fatty-acid biosynthesis), and energy metabolism (nitrogen metabolism). These data suggested a crucial role of *cdc42* in an array of cellular processes and events of *B. bassiana*.

### 3.6. Link of Cdc42 to Pyruvate Metabolism and Related Pathways

The KEGG analysis revealed links of nine pathways to pyruvate metabolism. As illustrated in Figure 5, six identified DEGs involved in the pathway of pyruvate metabolism could affect pyruvate biosynthesis through the routes of glycine, serine, and threonine metabolism (BBA_00790 and BBA_04664), glycolysis/gluconeogenesis (BBA_04543 and BBA_08227), and ascorbate and aldarate metabolism (BBA_04538 and BBA_08261). Among those DEGs, five were upregulated while only one was downregulated (BBA_04664), implying that pyruvate biosynthesis could be induced in counteracting the effect of deleted *cdc42*. Moreover, two genes (BBA_08386 and BBA_08900) encoding pyruvate decarboxylase, which enables catalysis of pyruvate to acetaldehyde, were downregulated, suggesting reduced pyruvate consumption in the absence of *cdc42*. These dysregulated genes implicated an increase in pyruvate accumulation in the Δ*cdc42* mutant. Moreover, several upregulated genes (BBA_00790, BBA_09252, BBA_01687, BBA_04309, and BBA_08608) were involved in the participation of pyruvate in tricarboxylic acid (TCA) cycle and/or valine, leucine, and isoleucine biosynthesis. Three other upregulated genes (BBA_ 08607, BBA_02336, and BBA_07304) were also involved in the TCA cycle. In addition, most DEGs involved in fatty-acid biosynthesis, glycerophospholipid metabolism, ether lipid metabolism, and arginine biosynthesis were differentially upregulated. These transcriptomic data suggested a novel role for Cdc42 in the pyruvate metabolism and related pathways of *B. bassiana*.

### 3.7. Validated Role of Cdc42 in Pyruvate Metabolism, TCA Cycle, and Virulence Maintenance

The novel role of Cdc42 suggested by transcriptomic analysis was clarified by quantification of pyruvate contents and TCA cycle-related enzyme activities and ATP levels from the extracts of 5-day-old SDBY cultures generated by shaking incubation at 25 °C. As a result, intracellular pyruvate content was enhanced by 22% in Δ*cdc42* relative to the WT strain (Figure 6A), coinciding well with the roles of those identified genes in pyruvate metabolism. The TCA cycle crucial for the catalysis of pyruvate into other metabolites and energy production was also revealed by transcriptomic analysis to be accelerated by upregulated expression of ATP-citrate synthase subunit 1 (BBA_8608), Succinyl-CoA synthetase-like protein (BBA_8607), fumarate hydratase (BBA_2336), and malate dehydrogenase (BBA_7304). The accelerated TCA cycle was proven by an increase in citrate synthase activity by 6% and in fumarate hydratase by 27% (Figure 6B), as well as an elevation in ATP level by 16% (Figure 6C) in Δ*cdc42* compared to the WT strain.

Moreover, transcriptomic analysis revealed the involvement of many DEGs in secondary metabolisms. Beauvericin is a secondary metabolite considered as an important virulence factor in *B. bassiana* [9], and its production was revealed under the influences of downregulated genes (BBA_02942, BBA_02945, and BBA_06422) identified in the transcriptome. Revealed by HPLC, the contents of beauvericin in the supernatants of the 5-day-old SDBY cultures were averagely lowered by 50% in Δ*cdc42* versus the WT strain (Figure 6D). In addition, secreted Pr1-family proteases are collectively required for cuticle degradation during the normal infection [7]. In the present study, the total activities of Pr1 protease assessed from the supernatants of the 5-day-old SDBY cultures on average decreased by 19% in Δ*cdc42* in comparison to the WT strain (Figure 6E). The two measurements demonstrated an important role for Cdc42 in host infection and insecticidal activity of *B. bassiana*, well in accordance with blocked cuticle infection and greatly attenuated virulence observed in the Δ*cdc42* mutant.

## 4. Discussion

In model yeast, intensive studies have focused mainly on a regulatory role of Cdc42 in the cell cycle and/or polarity establishment [19,32]. Our study unravels not only a conserved role of Cdc42 in the cell cycle and hyphal septation pattern, but also its novel role in the pyruvate metabolism and related pathways associated with the TCA cycle in *B. bassiana*. This novel role was revealed by multiple dysregulated genes and clarified by the alterations of intracellular pyruvate accumulation, related enzyme activities, and ATP levels in the absence of *cdc42*. These findings help to understand an indispensable role of Cdc42 in sustaining the fungal insect-pathogenic lifestyle and insecticidal activity, as discussed below.

Previously, an importance of orthologous Cdc42 for virulence was reported in different fungal pathogens [36,37,38,39]. Cdc42 is evidently required for host penetration and virulence as well as its effect on germination and sporulation in *Magnaporthe grisea* [47]. However, mechanisms underlying the regulatory role of Cdc42 in fungal virulence remain elusive. In the present study, the Δ*cdc42* mutant was severely compromised in a capability of infecting the model inset through the normal route of cuticular penetration and of colonizing the insect hemocoel by yeast-like budding proliferation in vivo. The compromised capability was obviously associated with those dysregulated genes identified in transcriptomic analysis. Importantly, the abnormal acceleration of pyruvate metabolism and TCA cycle-related pathways revealed by the analysis were validated by increased accumulation levels of intracellular pyruvate, ATP, and enzymes associated with the TCA cycle. This highlights a critical role for Cdc42 in the maintenance of a normal pyruvate metabolism and TCA cycle in *B. bassiana*. Pyruvate has been shown as a significant virulence factor in *Candida albicans* [48]. However, in *B. bassiana*, accumulation of pyruvate seems to be associated with the *cdc42*-dependent virulence depression. Due to the pathways dysregulated in the absence of *cdc42*, the decreased accumulation of extracellular beauvericin as a virulence factor [9] was concurrent with the reduced secretion of Pr1-family proteases vital for cuticle degradation [7]. The dysregulated pathways could also exert impacts on some other cellular processes and events, such as delayed germination, retarded hyphal growth, and reduced tolerance to heat and UV-B irradiation. Therefore, we infer that an indispensability of Cdc42 for virulence is an overall output of multiple cellular processes and events associated with its regulatory role in the pyruvate metabolism and related pathways in *B. bassiana.*

Moreover, the conserved role of Cdc42 elucidated in model yeast [19,32] is somewhat differentiated in *B. bassiana*. Our microscopic examination of germinating conidia and their germ tubes revealed inconspicuous alteration in polar growth on agar plates when *cdc42* lost function. Instead, a disturbed cell cycle occurred in the mutant’s blastospores produced in the submerged SDBY cultures, accompanied by the formation of multicellular hyphae with reduced or increased septa. The increased septa resulted in the formation of short stout cells markedly altered in morphology. Since the submerged culture in vitro is mimic to the yeast-like budding proliferation in the insect hemolymph, we speculate that in *B. bassiana*, Cdc42 could have evolved towards the fungal adaptation to the insect-pathogenic lifecycle. This speculation also gives an explanation for a great loss of the mutant’s virulence to the model insect.

In conclusion, Cdc42 is indispensable for the insect pathogenicity and insecticidal activity of *B. bassiana*. This indispensability relies upon its regulatory role in the normal expression of clustered genes, particularly those involved in the pyruvate metabolism and TCA cycle-related pathways. Therefore, we unveiled the function of Cdc42 in the fungal cell cycle and insect-pathogenic lifestyle in *B. bassiana*. In addition, transcriptomic analysis and experimental data support the hypothesis that pyruvate metabolism and energy production was influenced by Cdc42 and associated with the fungal biological potential. These findings not only offer a novel insight into a regulatory role of Cdc42 in filamentous fungal pathogens, but also provide new ideas for optimizing microbial pesticides and their application potential.

## Figures and Tables

**Figure 1 jof-08-00394-f001:**
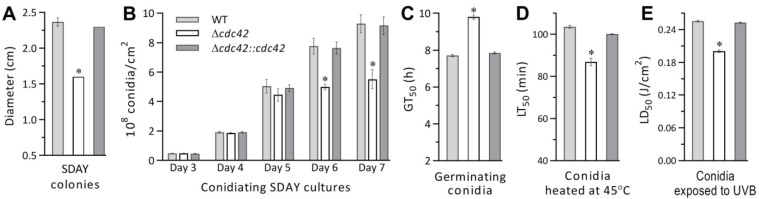
Role of *cdc42* in the asexual cycle in vitro of *B. bassiana*. (**A**) Diameters of SDAY colonies incubated for 7 days after initiated with ~10^3^ conidia at the optimal regime. (**B**) Conidial yields measured from the SDAY cultures during a 7 day incubation after initiated by spreading 100 μL aliquots of a 10^7^ conidia/mL suspension at the optimal regime. (**C**–**E**) GT_50_ (h), LT_50_ (min), and LD_50_ (J/cm^2^) estimates as the respective indices of conidial quality for the time length of 50% germination at 25 °C, tolerance to a wet-heat stress at 45 °C and resistance to UVB irradiation. * *p* < 0.05 in Tukey’s HSD tests. Error bars: SDs from three replicates.

**Figure 2 jof-08-00394-f002:**
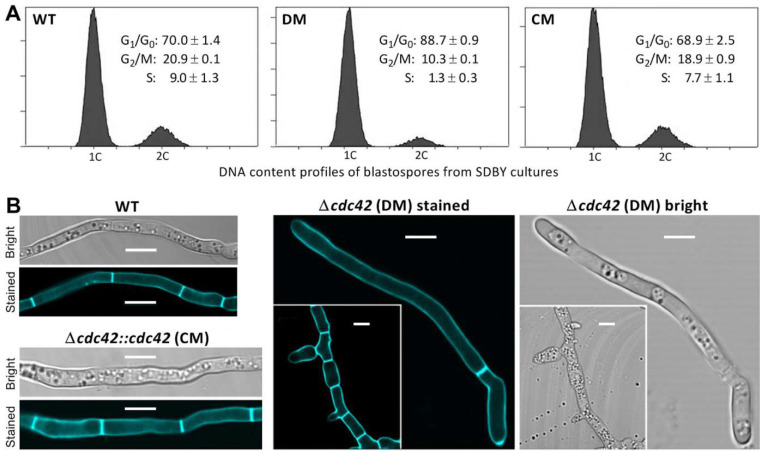
Impact of *cdc42* deletion on the cell cycle of blastospores and the formation of hyphal septa in *B. bassiana*. (**A**) Cell-cycle patterns of unicellular blastospores stained with the DNA-specific dye propidium iodide after collection from the 3-day-old SDBY colonies incubated at optimal 25 °C. Different cell-cycle phases are shown as standard deviations of the means from three samples per strain (2 × 10^4^ stained spores per sample). (**B**) LSCM images (scales: 5 μm) of hyphal septum patterns and morphology. The presented hyphae were stained with the cell-wall-specific dye calcofluor white after collection from the 3-day-old SDBY cultures. Note that two opposite types of abnormal septum patterns are present in the Δ*cdc42* mutant’s culture.

**Figure 3 jof-08-00394-f003:**
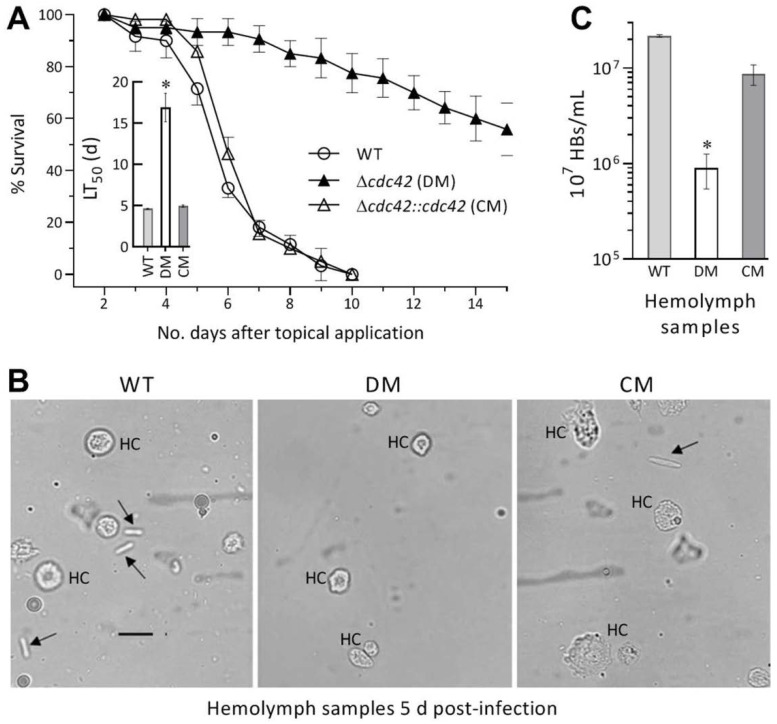
Essential role of *cdc42* in insect-pathogenic lifestyle of *B. bassiana*. (**A**) Survival percentages of *G. mellonella* larvae after topical application (immersion) of a 10^7^ conidia/mL suspension for normal cuticle infection and LT_50_ (d) estimates made by modeling analysis of time-mortality trends. (**B**) Microscopic images (scale: 20 μm) for the status of hyphal bodies (arrowed) and insect hemocytes (HC) in the hemolymph samples taken from surviving larvae 5 d post-infection. (**C**) Concentrations of hyphal bodies (HBs) from the hemolymph samples. * *p* < 0.05 in Tukey’s HSD tests. Error bars: SDs from three independent replicates (**A**) or three surviving larvae infected per strain (**C**).

**Figure 4 jof-08-00394-f004:**
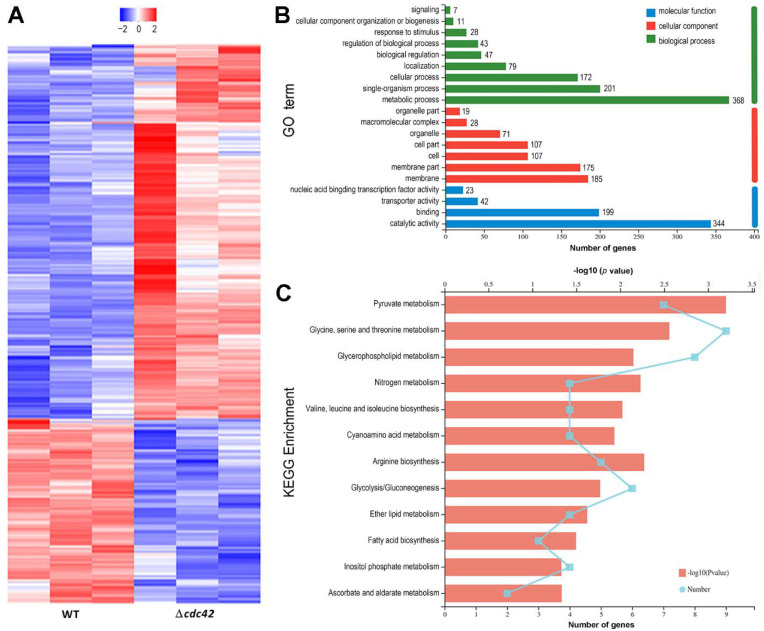
Transcriptomic analysis of Δ*cdc42* versus WT strains in *B. bassiana*. (**A**) Heat map of differentially expressed genes (DEGs). (**B**) Counts of DEGs enriched to GO terms of three function classes. (**C**) Counts of DEGs enriched to KEGG pathways at different levels of *p* values.

**Figure 5 jof-08-00394-f005:**
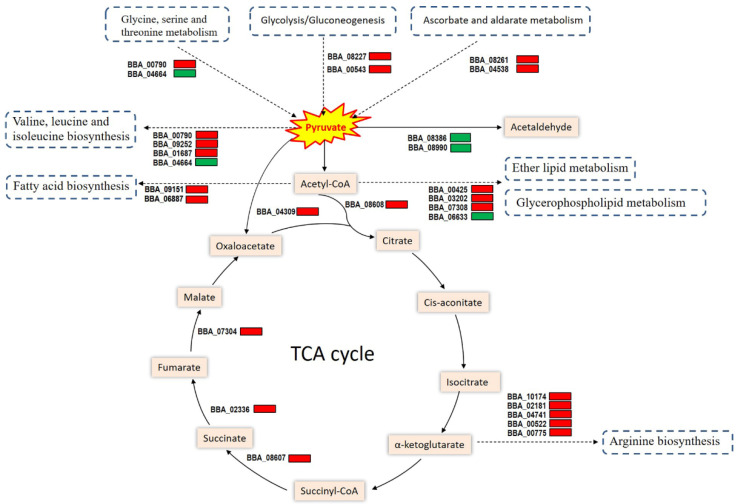
Diagram for involvements of identified DEGs in the pyruvate metabolism and related pathways of the Δ*cdc42* mutant versus the WT strain. Red and green denotes up- and downregulated genes, respectively. Each gene with its tag locus in the *B. bassiana* genome is illustrated with its expression status in Δ*cdc42* (red or green) as compared to WT.

**Figure 6 jof-08-00394-f006:**
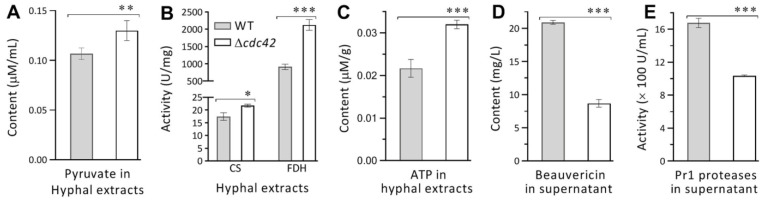
Impacts of *cdc42* deletion on pyruvate metabolism, energy production, and virulence-related beauvericin and Pr1 proteases in *B. bassiana*. (**A**–**C**) Pyruvate contents, activities of TCA cycle-related citrate synthase (CS) and fumarate dehydrogenase (FDH), and ATP contents quantified from the extracts of 5-day-old SDBY cultures incubated at 25 °C, respectively. (**D**) Beauvericin contents measured from the supernatants of the 5-day-old SDBY cultures by HPLC. (**E**) Total activities of Pr1 family proteases quantified from the supernatants of the 5-day-old SDBY cultures. *p* < 0.05 *, 0.01 ** or 0.001 *** in Student’s *t* tests. Error bars: SDs from three independent samples analyzed.

## Data Availability

Transcriptomic (RNA-seq) data analyzed in this study are openly available in the Sequence Read Archive (SRA) with the accession number of PRJNA818182.

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
