# Peer review of "Unveiling a Novel Role of Cdc42 in Pyruvate Metabolism Pathway to Mediate Insecticidal Activity of Beauveria bassiana"

_jof, 2022, doi:10.3390/jof8040394_

Round 1
Reviewer 1 Report
ABSTRACT
- Line 3. I suggest changing "an indispensability" for "the indispensability"
- Line 5. Please mention the model insect
- Penultimate late: please write in full "tricarboxylic acid cycle (TCA)"
MATERIALS AND METHODS
- Line 76: Explain "normal growth"
- Line 141: what means "(d)"?
- Line 156. Please detail more the parameters used in the collection of data in flow cytometer.
- Also, please check the first mention of TCA in the text, in order to write in full.
RESULTS
- Line 226: "regime of"
- Line 230: "was also"
- Line 243: "important role"
- Line 249: Why these data are not being showed?
CONCLUSION
- Lines 422-423. Please elaborate more this last sentence.
- What are the potential applications / offshoots of the research?
GENERAL COMMENT
- Please check the manuscript to correct minor language/spell problems.
Reviewer 2 Report
General comments
- The manuscript is suitable for publication after some minor corrections
- The scientific approach is good; the subject is up to date and informative.
- The paper is well written, and the objectives of the study are clearly stated.
- The paper is well discussed.
Special comments
- Specific comments have been made in the text

Reviewer 3 Report
The study reports on targeted disruption of the cdc42 gene encoding a Rho family small GTPase in the entomopathogenic fungus Beauveria bassiana and investigation of related transcriptomic and phenotypic effects in comparison to the wild type and a rescue mutant. Several groups of differentially expressed genes and several groups of altered phenotypic traits were identified. Most prominent traits concern cell cycle/hyphal growth, virulence on laboratory bioassays against Galleria mellonella larvae/virulence factor production and changes in pyruvate metabolism/TCA cycle. Experiments appear very well done and presentation is generally very appropriate and clear.
Remarks of minor importance:
1) Throughout the manuscript there is an obvious problem with blanks missing, leading to word fusions (e.g. line 105). Please check also for typos: "also" (line 230), "changes" (242), "cultures"(251), "counterparts" (252), "residued" (253), "metabolism" (318), "averagely" (362), "proteases" (365), "insect" (393), some more.
2) Agrobacterium-mediated transformation of Beauveria might not appear a standard technique to many readers. It might be appropriate to say some respective words in the introduction section.
3) The study is based on B. bassiana strain ARSEF 2860. The Introduction section or Section 2.1 might reasonably be enriched by some information on this strain: why has it been chosen for disruption experiments (model organism, application relevance, …), where does it stem from (geographic origin, original host), has it been used in previous relevant published research?
4) Legend to Figure 5: “Each gene with its tag locus in the B. bassiana genome is illustrated with its ex- 341 pression status in WT (white) and Δcdc42 (red or green)”. As WT levels are used to normalize comparative expression data, the “expression status” indication for the WT does not correlate to the expression status of the respective gene. It does not contain any informational content. By definition always white rectangles could simply be removed from Figure 5.
With respect to the interpretation of experimental results, authors’ argument starts from the assumption that pathogenicity has been demonstrated to be cdc42 ortholog dependent in other fungal pathogens, but that the “mechanism underlying the regulatory role of Cdc42 in fungal virulence” remains unelucidated (lines 388ff).
The present study confirms cdc42 dependence of insecticidal activity for Beauveria bassiana. Moreover, cdc42 disruption is found to cause transcriptional dysregulation of large gene sets, and acceleration of pyruvate metabolism is well demonstrated to be cause of cdc42 disruption.
It must be in line with expectations that disruption of a small GTPase gene as cdc42 will cause broad transcriptional dysregulation and numerous pleiotropic effects. However, it appears the main message of the paper that the role of cdc42 in insecticidal activity of the WT is “mediated” (title) by its role in pyruvate metabolism.
With respect to the regulatory mechanism, the authors refer to “an overall output of multiple cellular processes and events associated with the regulatory role (of Cdc42) in the pyruvate metabolism and related pathways”. This appears to say that as pyruvate metabolism/TCA is central for cellular energy metabolism, its disturbance will affect next to every energy dependent process across the cell or organism, including virulence.
At this level, the conclusion presented appears ambiguous. It is, of course, in line with ex ante assumptions that dysregulation of the pyruvate metabolism will affect overall organismal fitness, and that virulence as a general fitness dependent trait might in turn be impaired. Is it substantially this what the authors want to state – corroborating their claim by excellent experimental data that replace ex ante assumptions? Or do they want to exclude the existence of a more specific regulatory mechanism as, e.g., a signaling cascade linking cdc42 with entomopathogenic fungal virulence factors by-passing pyruvate metabolism regulation? Presentation of the conclusion might be more explicit at this level.
Moreover, it is irritating that with respect to the conclusion drawn the discussion section hardly exceeds the own data presented. Have there not been previous studies that could independently corroborate supposed effects of the numerous traits claimed here to be cdc42-dependent (pyruvate or ATP accumulation) upon entomopathogenic fungal virulence or virulence factor expression? The authors might feel invited to discuss their conclusion in a wider scientific context.
